# A genomics learning framework for undergraduates

**Laura K. Reed**[1], **Adam J. Kleinschmit**[2], **Vincent Buonaccorsi**[3†], **Arthur G. Hunt**[4], **Douglas Chalker**[5], **Jason Williams**[6], **Christopher J. Jones**[7], **Juan-Carlos Martinez-Cruzado**[8], **Anne Rosenwald**[9]*

1 Department of Biology, University of Alabama, Tuscaloosa, Alabama, United States of America, 2 Department of Natural and Applied Sciences, University of Dubuque, Dubuque, Iowa, United States of America, 3 Department of Biology, Juniata College, Huntingdon, Pennsylvania, United States of America, 4 Department of Plant and Soil Sciences, University of Kentucky, Lexington, Kentucky, United States of America, 5 Department of Biology, Washington University in St. Louis, St. Louis, Missouri, United States of America, 6 DNA Learning Center, Cold Spring Harbor Laboratory, Cold Spring Harbor, New York, United States of America, 7 Department of Biological Sciences, Moravian University, Bethlehem, Pennsylvania, United States of America, 8 Department of Biology University of Puerto Rico at Mayagüez, Mayagüez, Puerto Rico, United States of America, 9 Department of Biology, Georgetown University, Washington, DC, United States of America

† Deceased.

* anne.rosenwald@georgetown.edu

**Data Availability Statement:** All relevant data are within the manuscript and its Supporting information files.

**Funding:** The author(s) received no specific funding for this work.

## Abstract

Genomics is an increasingly important part of biology research. However, educating undergraduates in genomics is not yet a standard part of life sciences curricula. We believe this is, in part, due to a lack of standard concepts for the teaching of genomics. To address this deficit, the members of the Genomics Education Alliance created a set of genomics concepts that was then further refined by input from a community of undergraduate educators who engage in genomics instruction. The final genomics concepts list was compared to existing learning frameworks, including the Vision and Change initiative (V&C), as well as ones for genetics, biochemistry and molecular biology, and bioinformatics. Our results demonstrate that the new genomics framework fills a niche not addressed by previous inventories. This new framework should be useful to educators seeking to design stand-alone courses in genomics as well as those seeking to incorporate genomics into existing coursework. Future work will involve designing curriculum and assessments to go along with this genomics learning framework.

## Introduction

Genomics is the study of the genome of a single organism or a set of organisms and may include organelle genomes (e.g., mitochondrial, chloroplast, and apicoplast). Genomics as a field has been facilitated by technological advances in genome sequencing (see ref. [1] for historical context). While much has been learned about genes and the relationship between genotype and phenotype, genomics provides a more complete understanding of the polygenic nature of traits and the role that evolution plays in the development of variation, including

**Competing interests:** The authors have declared that no competing interests exist.

speciation. The field of genomics is characterized by concepts derived from molecular biology and evolution but derives conclusions from analysis of entire genomes. Genomes include not only the sequences coding for proteins and RNA products ("genes") but also regulatory sequences. In the case of eukaryotes, much of the genome can be made up of repetitive elements, including transposable elements and their remnants. Repetitive elements have a major impact on both genetic regulation and evolution by affecting DNA packaging. In contrast, prokaryotic genomes are typically minimal, having been stripped down to essential information by selection. In the learning framework presented here, "genome" will mean this complex assembly of coding and non-coding information.

Genomics plays an increasingly valuable part of biological research. For example, a set of 27 papers was published in 2020 that examined the genomes of several thousand different cancers compared to the corresponding normal tissues (the set is described in the flagship paper, ref. [2]). As another example, genomics is increasingly used to evaluate biodiversity and to mitigate biodiversity loss due to habitat loss and climate change [3]. Given this increased use of genomics for research, it is critical that life science undergraduates become familiar with the tenets of genomics so that they are better equipped for graduate study and entry into the workforce. Both academic and applied research fields are increasingly reliant on aspects of data science, and genomics provides a good introduction to "big data". Indeed, genomics offers wonderful opportunities for students to engage in authentic, hands-on research [4–6] that can be undertaken anywhere with, at minimum, a computer and an internet connection, allowing more students to become involved in research than is possible with laboratory-based apprenticeship models. Examples of classroom-based undergraduate genomics research programs include the Genomics Education Partnership [7–10], GCAT-Seek [11], Genome Solver [12], the Ciliate Genomics Consortium [13, 14], SEA-PHAGES [15, 16] among others. There are numerous places where faculty can find curriculum grounded in genomics concepts including the Genomics Education Alliance (https://qubeshub.org/community/groups/gea/), the NIBLSE Resource Collection (https://qubeshub.org/community/groups/niblse/), the *CourseSource* Bioinformatics course (https://qubeshub.org/community/groups/coursesource/), and the Cold Spring Harbor Laboratory DNA Learning Center (https://dnalc.cshl.edu), including the DNA Subway platform (http://www.dnasubway.org).

With the simplification and miniaturization of sequencing technologies and decreases in costs, hands-on genomics increasingly happens in the classroom. A hand-held Oxford Nanopore MinION device generates up to 50 billion DNA bases–the equivalent of 16 human genomes–in 72 hours for less than $1500. Nanopore startup costs are 30–50 times less than other high-throughput sequencing platforms. Nanopore's mobility, ability to generate long reads (>1 million bases), and increasing accuracy have resulted in its adoption for *de novo* genome assembly, detection of DNA modifications, RNA sequencing, and metagenomics [17, 18]. The DNA Learning Center, for example, routinely offers Nanopore sequencing summer courses for high school students (https://summercamps.dnalc.org/camps/sequence-a-genome.html).

An overall framework for improving life sciences education, "Vision and Change in Undergraduate Biology Education" (V&C), was developed as a project among numerous stakeholders, including the National Science Foundation and the American Association for the Advancement of Science [19, 20]. While providing a critical starting point, the V&C concepts for biological literacy are broadly descriptive, and although many of the concepts can be illustrated through genomics-related curricula, specific concepts for genomics have not been elaborated like in other life science subdisciplines, including microbiology [21], biochemistry and molecular biology [22], general biology [23–25], immunology [26], neuroscience [27], ecology (https://www.esa.org/4dee/framework/), toxicology [28], pharmacology [29], and

bioinformatics [30]. To assist undergraduate educators in designing and implementing a curriculum in genomics, we have identified a collection of key concepts we view as central to an understanding of genomics.

Genomics can be seen as an extension of and overlapping with genetics, molecular biology and/or biochemistry. One of the first concept inventories for genetics was elaborated by Hott and colleagues [31] and an assessment instrument using these concepts was developed by Bowling and colleagues [32]. Another related assessment was developed by Smith, Wood, and Knight [33], then used to assess literacy in genetics [34], with a revision of the learning framework in 2015 [35]. There are several overlapping inventories of concepts and skills for biochemistry and molecular biology, including those by Tansey and colleagues [22], Wright and Hamilton [36], and White and colleagues [37]. Interestingly, Loertscher and colleagues [38] elaborated a set of threshold concepts for biochemistry, ones that "when mastered, represent a transformed and integrative understanding of a discipline without which the learner cannot progress." Some molecular biology concept inventories are focused on specific topics such as the Central Dogma [39] and meiosis [40].

Similarly, competencies and concept inventories have been elaborated for bioinformatics, a field also closely aligned with genomics. While genomics focuses on the sequence information contained within an organism or set of organisms and its meaning, bioinformatics describes the analysis of such information and the tools that can be applied to the study of genomic data. Recently, a group of undergraduate educators developed a set of competencies; these competencies were then refined by a national survey of life sciences faculty to a set of nine [30]. In addition, a mastery rubric for bioinformatics was developed by Tractenberg and colleagues [41].

However, to our knowledge, no similar framework for genomics specifically has been developed. A search of the literature was conducted at the beginning and the end of our development of the genomics concepts. Although there are disciplinary genomics concept inventories, primarily for healthcare professionals, including, for example, genetics and genomics competencies for nurses [42, 43], none exists for life sciences undergraduates. We therefore attempted to rectify this situation by developing a set of genomics concepts for this population of students.

## Methods

Members of the Genomics Education Alliance (GEA; https://qubeshub.org/community/groups/gea) developed the initial genomics concepts in 2019 and 2020, seeded by the learning outcomes of one member's advanced genomics course, and enhanced with *ad hoc* feedback from 20 fellow genomics educators from a combination of 2-year, 4 -year, and research-intensive institutions, both public and private, all from the United States. In 2021, additional faculty who teach genomics to undergraduates were surveyed to assess both the specific content of the initial concepts (S1 File) and to determine the relative importance of the concepts to genomics education at various levels (e.g., introductory versus advanced biology majors). The survey was completely anonymous and there was no incentive for participation. Survey participants were recruited through social media (e.g., Twitter), targeted emails to relevant professional groups including the Genomics Education Partnership, Genome Solver, GCAT-Seek, and by asking respondents to send to their contacts (snowball recruiting). The survey was available from 1/14/2021 to 5/24/2021. Consent was implied by the completion of the survey. Sixty-one individuals completed the survey. The survey data was analyzed and reported under IRB protocols at Georgetown (#00006349) and Washington University in St. Louis (#201809064), and deemed to be exempt.

The survey included basic information about the context in which the faculty member teaches genomics (e.g., class size, level of students) and then a Likert scale evaluation of the "importance" (1 = not important, 5 = extremely important) of each concept to undergraduate biology majors' education. Participants were then asked which of the concepts they include in their own teaching. Finally, participants were asked to provide feedback on the content of the concepts and identify any additional areas that might need to be included. Summary statistics were calculated from the survey results, as well as basic analysis of any correlations between the faculty member's teaching context and their responses about the importance of a given concept.

The initial framework (S1 Table) was then refined based on the feedback from the faculty community and by the GEA members (n = 17). After feedback and refinement, an additional concept category was added to address the ethical and social implications of genomic research and technology.

We then compared our concept list to established inventories in several related fields, as well as V&C [19] using degrees of freedom analysis [44], which allows for alignment of two inventories at a time. These inventories included genetics inventories, those by Smith, Wood, and Knight [33] and the CourseSource Genetics Learning Framework (https://qubeshub.org/community/groups/coursesource/courses/genetics); biochemistry and molecular biology, those by Tansey *et al.* [22], Loertscher *et al.* [38] and the CourseSource Learning Framework for Biochemistry and Molecular Biology (https://qubeshub.org/community/groups/coursesource/courses/biochemistry-and-molecular-biology); and bioinformatics, those by Wilson Sayres *et al.* [30], Tractenberg *et al.* [41], and the CourseSource Bioinformatics Learning Framework (https://qubeshub.org/community/groups/coursesource/courses/bioinformatics). Two GEA members independently identified the conceptual overlap between the proposed Genomics Concepts and the other existing inventories. Any differences in overlap assignment were then reconciled through discussion.

## Results

### Development of the concepts

The Genomics Education Alliance (GEA; https://qubeshub.org/community/groups/gea) was established in 2018 by several undergraduate faculty who had developed hands-on genomics research projects. During discussions about how to best assess genomics learning outcomes, the participants realized that the field lacked an appropriate framework for genomics. Discussions among faculty as well as searches through the literature revealed a lack of agreed-upon genomics concepts. We therefore undertook the creation of a tool for educators.

The initial set of concepts was generated by one faculty member from their Advanced Genomics course, then discussed informally among members of GEA (S1 Table). We also sought input from attendees at the 2020 BIOME Summer Institute (n = 15) [45]. The inventory was intended to collect the set of genomics-related concepts that an undergraduate biology major should understand by the time they graduate. Initially, the concepts were partitioned into five Biological Concepts—three with subheadings (what the genome is and how it functions), and seven Methodological Concepts (how one studies genomes and genomic data) (S1 Table).

This initial set (outlined in S1 Table) was then refined by input via a survey (S1 File) from a broader group of undergraduate biology educators incorporating genomics concepts into their teaching. Sixty-one individuals engaged in genomics education at the college level completed the survey. Of those 61 individuals 30 were or had been engaged in teaching genomics concepts at both the introductory and upper levels, 20 only taught genomics at the upper level, six

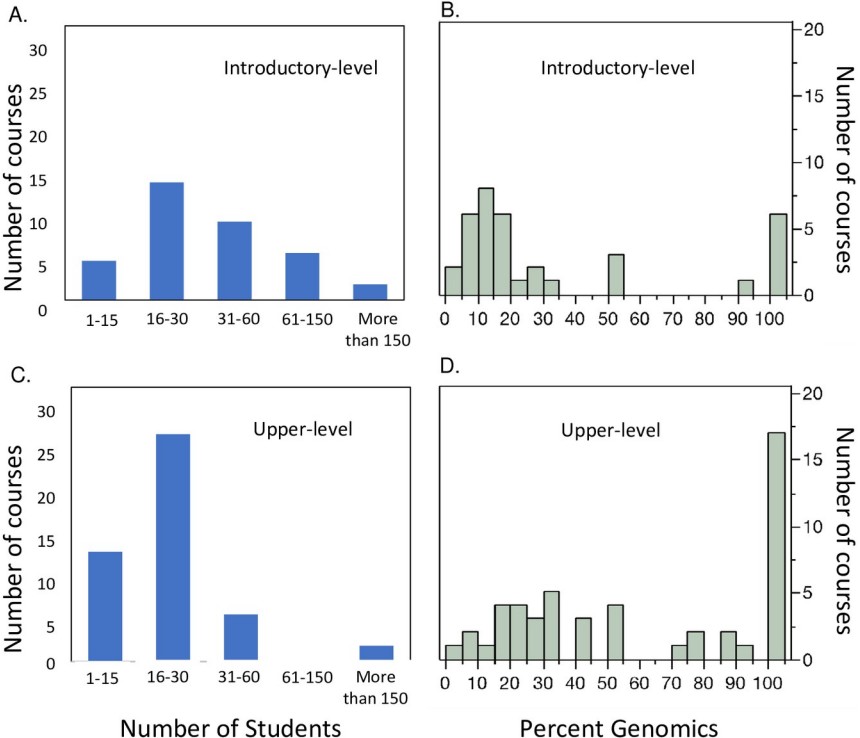

**Fig 1. Teaching context of faculty participating in the inventory survey.** Class size (A) and percent of course devoted to genomics concepts (B) in introductory (first and second year) courses. Class size (C) and percent of course devoted to genomics concepts (D) in upper-level (third and fourth year) courses.

taught genomics only at the introductory level, and five were not teaching genomics at the time of the survey. At the introductory level, faculty were more likely to be teaching large enrollment classes with less emphasis on genomics. The proportion of the course focused on genomics tended to increase in upper-level classes (Fig 1). We also found that the courses with smaller class sizes tended to include a greater focus on genomics at both the introductory and upper level (S1 Fig).

The faculty indicated the importance of the initially proposed concepts for biology majors to learn. All of the initial concepts were evaluated to be moderately to highly important on average (>3.73 on a 5-point scale, S2 & S3 Figs, S2 Table). Overall, the Biological Concepts (4.29) were rated as more important than the Methodological Concepts (4.17, p<0.011, by Least Squares ANOVA). The four most highly ranked concepts included B.A (4.64, common ancestry), B.D.2 (4.70, central dogma), B.E (4.64, sequence variation) and B.E.6 (4.59, ethics), and the three least important concepts were B.E.3 (3.74, transposable elements) and B.E.5 (3.80, virus genomes), and Methodological Concept M.E (3.89, metagenomics).

We asked the survey respondents whether they taught these concepts in their introductory and upper-level courses. All concepts were taught in at least some courses. Two of the initial Biological Concepts were taught in a significantly higher proportion of advanced relative to introductory courses (B.B & B.D), while all of the Methodological Concepts (except M.G) were taught in a significantly larger proportion of upper-level courses (S4 Fig). Overall, there was also a higher probability that faculty members taught the biological concepts than the methodology concepts (p = 0.003, t-test). Not surprisingly, the probability that a faculty member taught a given concept, in either an introductory or upper-level course, was positively

correlated with how "important" the faculty member judged that concept to be (p<0.0001, t-test). Based on the feedback from the survey, the concepts were modified to include new ideas and reorganized into a more logical structure. One major addition, based on strong feedback from the community (10% of respondents), was the creation of a new category, "Ethical Concepts" (ECs), to elevate the importance of teaching ethics as part of genomics (initially listed as biological subconcept B.E.6). Within this category two overarching concepts were developed, one to focus on the impact of genomics on human well-being directly (EC.I and subparts), and the second to focus on genomic impacts on the environment (EC.II and subparts). Biological Concepts about haplotypes (BC.II.B), pangenomes (BC.II.G), and elaboration on cellular genomes of multicellular organisms (BC.V, BC.V.A-BC.V.D) were also added based on community feedback. Finally, based on survey comments, Methodological Concepts about systems biology (MC.V.D), biotechnology (MC.VI, MC.VI.A, MC.VI.B), and elaboration on genomics as a data science (MC.VII.A, MC.VII.B) were added. The final product, an integrated and comprehensive set of core genomics concepts that can guide undergraduate curriculum development, is presented in Table 1. The mapping of the initial concepts sent for survey feedback and of the finalized concepts described in Table 1 is shown in S2 Table.

## Comparisons to existing learning frameworks

In addition to developing the set of Genomics Concepts reported in Table 1, we also determined the extent to which our genomics concepts captured new information by comparing the new inventory to existing inventories for genetics ([33], https://qubeshub.org/community/groups/coursesource/courses/genetics), biochemistry and molecular biology ([22], [38], https://qubeshub.org/community/groups/coursesource/courses/biochemistry-and-molecular-biology), bioinformatics ([30], [41] https://qubeshub.org/community/groups/coursesource/courses/bioinformatics) as well as V&C ([19]; Table 2). We used a degrees-of-freedom analysis, previously described in the work of Tractenberg [44], in which we constructed grids with the Genomics Concepts as a column and the elements of the inventory to compare to as a row across the top. Each comparison was done independently by two of the authors (LKR and AR) and then compared to each other. Discrepancies were resolved by discussion. Rationale for the assignment of a "match" is given in the appropriate cell of the alignment table.

As an example, we show in Table 2 the reconciled alignment between the genomics concepts and the V&C goals. The V&C goals were designed to elaborate aims for biology education broadly (beyond just genomics, in other words), but nevertheless a number of the genomics concepts resonated with these goals, including an understanding of evolution as one of the core concepts and the use of statistics and modeling as one of the core competencies. We note that all of the V&C Concepts and Competencies are represented in at least one of the genomics concepts and that the genomics concepts are represented in at least one of the V&C Concepts and Competencies, suggesting that the genomics concepts are broadly useful as a way to implement the goals of V&C.

Similarly, we compared the genomics concepts to inventories for disciplines related to genomics. As expected, narrowly focused concept inventories had little overlap with the Genomics Concepts (*i.e.*, the Meiosis Concept Inventory [40], the Central Dogma Concept Inventory [39]). On the other hand, genetics concepts showed relatively large overlaps with the genomics concepts (**Supplemental 3 and 4**), although at least one of the genetics concept inventories [33] is focused primarily on eukaryotic organisms, including humans, with greater concerns about the effects of mutation on human health than we represent in the genomics concepts. Biochemistry/molecular biology inventories showed intermediate levels of overlap, consistent with the different emphases in this discipline on protein structure and function,

**Table 1. Final list of genomics concepts.** The list of genomics concepts shown here was developed through several iterations of community input from undergraduate life sciences educators who teach genomics. The intended audience for this list is other undergraduate educators seeking to develop courses in genomics or modify existing courses to contain more genomics elements.

**Biological Concepts (BC)**

BC.I. Evolutionary forces and processes inform our understanding and interpretation of genomic information.

| | |
|---|---|
| BC.I.A. Phylogenetic relationships, reflecting common ancestry, can both be determined by and facilitate interpretation of genomic data. | BC.I.A.1. Comparative genomics can provide new insights into structure and function of genes, and provides insight into evolutionary forces acting on processes such as de novo gene birth, gene duplication, and sub-functionalization. |
| | BC.I.A.2. Genomic traits can be used to define evolutionary relationships across phylogenies, including evolutionary mechanisms such as vertical inheritance and horizontal gene transfer. |
| BC.I.B. Understanding the evolutionary history of the population and species under study informs interpretation of genomics information. | BC.I.B.1. Evolutionary forces such as mutation, recombination, selection, demography, migration, admixture, and genetic drift (neutral evolution) influence properties and patterns of genome sequence and organization within and between species. |
| BC.I.C. Evolution of genome structure includes changes in chromosome or genome copy number, genomic rearrangements, conservation of or change in gene order (local synteny), gene duplication or loss, and horizontal gene transfer. | BC.I.C.1. Meiotic processes in eukaryotes such as recombination and genome replication contribute to structural evolution. |
| | BC.I.C.2. Changes in genome structure such as copy number variation, aneuploidy, and whole genome duplication contribute to organism phenotypes including human disease risk. |

BC.II. Genomes exhibit sequence variation (SNPs, etc.) and structural variation (rearrangements, copy number variation, presence/absence of transposable elements) within and between species that can lead to differences in form and function.

BC.II.A. Phenotype is in part determined by patterns of gene expression that vary between individuals and between species as a result of environmental cues.

BC.II.B. Haplotypes result from linkage disequilibrium between loci within chromosomes.

BC.II.C. Not all molecular features (e.g., introns, promoters, stop codons, genetic codes) of genes and genomes are common to all species or organelles.

BC.II.D. Eukaryotic genomes are distinctive in their capacity to retain both active copies and remnants of transposable elements (TEs), which can be the majority of the DNA, and can play a significant role in genome evolution.

BC.II.E. Eukaryotic genomes include nuclear and cytoplasmic (organellar) components

BC.II.F. Prokaryotic and eukaryotic genomes may (or can) acquire genes as a result of horizontal gene transfer, which provides drivers for evolution. Horizontal gene transfers can occur between organelles, commensal microbes, and their eukaryotic hosts.

BC.II.G. A species' pangenome consists of all the genes present across different strains of the same species, even if all strains do not carry all members of the pangenome.

BC.II.H. Viruses (including bacteriophages) have genomes that operate within their hosts and can have distinct evolutionary patterns due to their parasitic lifestyle. Viral genomes can have overlapping genetic elements.

BC.III. Individual regions of a genome (e.g., promoters, enhancers, protein coding regions, non-coding RNAs etc) can have different functions that are determined by sequence and structural features.

| | |
|---|---|
| BC.III.A. DNA exists in complexes with other molecules (largely proteins) that package it within the nucleus and/or cell and the physical organization of the genome is a major determinant of gene expression patterns. | BC.III.A.1. In eukaryotes, this packaging involves nucleosomes to form chromatin. |
| | BC.III.A.2. In prokaryotes, the nucleoid is a chromatin-like structure. |
| | BC.III.A.3. Functional states can be determined by epigenetics (DNA packaging into dynamic alternative chromatin states). |

BC.III.B. The three-dimensional structure of the genome is dynamic, varying through development, in response to environmental stimuli, and with the physiological state of the organism.

(*Continued*)

**Table 1.** (Continued)

| | |
|---|---|
| BC.III.C. Much of the genome does not code for proteins. | BC.III.C.1. Non-coding regions of the genome include origins of replication, promoters, enhancers, untranslated regions (UTRs), and non-coding RNAs. In addition, for eukaryotic genomes non-coding regions include centromeres, telomeres, repetitive DNA, and introns. |
| | BC.III.C.2. Promoters and enhancers influence when and how a protein-coding or non-coding RNA gene is expressed. UTRs of mRNAs also influence the expression on protein-coding genes. |
| | BC.III.C.3. The non-coding complement of eukaryotic genomes can vary by many orders of magnitude, such that some genomes may consist primarily of coding DNA, but others largely of non-coding DNA. |
| | BC.III.C.4. Much (even most) non-coding DNA may have no function. |
| | BC.III.C.5. Even functionless DNA can provide raw material for evolution, allowing spontaneous origination of new RNA- and protein-coding genes, novel gene regulatory mechanisms, and higher order genomic organization. |

BC.IV. Genomes contain information that determines both temporal and spatial patterns of gene expression and the response to environmental conditions.

BC.IV.A. Gene expression includes transcription of products that function as RNA molecules (tRNA, rRNA, lncRNA, miRNA, snoRNA, etc.).

| | |
|---|---|
| BC.IV.B. Expression of protein-coding genes includes transcription and processing to produce mRNA, followed by translation into protein. | BC.IV.B.1. For a given gene and its potential protein product, expression is regulated at the transcriptional, post-transcriptional, translational, and/or post-translational levels. This can include covalent modification of the product RNA and/or protein. |
| | BC.IV.B.2. In eukaryotes, alternative splicing is a means by which multiple proteins and mRNAs may be encoded by a single gene. |

BC.IV.C. Examination of the expression and evolution of all genes simultaneously can provide a deeper understanding of how genes are regulated and function both individually and in coordinated networks.

BC.V. Different cells in multicellular organisms can have different numbers of copies of the nuclear genome.

BC.V.A. Most somatic cells within a multicellular organism host a nearly identical copy of the diploid nuclear genome. Differentiation among somatic cells is largely due to changes in gene expression and not the genome.

BC.V.B. In sexually reproducing organisms, the diploid zygotic genome is recombined and assorted to generate haploid genomes for gametes.

BC.V.C. Some specialized cell-types within a multicellular organism are polyploid, containing many copies of the zygotic genome.

BC.V.D. Changes in zygotic genome ploidy through genome duplication can lead to speciation and evolutionary diversification.

BC.V.E. Mutations can accumulate and be selected within the genomes of somatic cells; these changes can lead to disease, including cancer, but cannot be inherited by subsequent generations through sexual reproduction.

BC.V.F. Mutations in the genomes of germline cells, those leading to gametes, can be inherited by subsequent generations through sexual reproduction.

BC.VI. The metagenome is composed of the community of microbial genomes and their expressed genes within a particular ecological niche (e.g., gut microbiome, soil)

BC.VI.A. The metagenome interacts across species, influencing the gene expression, metabolism, and phenotypes of all environmental community members.

BC.VI.B.—The hologenome arises from a symbiotic community (host and its symbiotic microbes) and co-evolves over evolutionary time.

**Methodological Concepts (MC)**

MC.I. Genome-scale studies have statistical and experimental design considerations that impact their accuracy.

(*Continued*)

**Table 1.** (Continued)

| | |
|---|---|
| MC.I.A. Consideration needs to be given to sample size, biological replication, technical replication, and study design. | |
| MC.I.B. There are specialized statistical metrics designed for genomic analyses. | |

MC.II. Within a species, genomic analyses provide information of how the genome shapes an organism's phenotype.

MC.II.A. Genomic sequencing and phasing of haplotypes (e.g., Genome Wide Association Studies, or GWAS) is a powerful tool for mapping genetic factors contributing to phenotypes including disease and for enhanced plant and animal breeding.

MC.II.B. Sequencing multiple individuals allows characterization of a population's prevalent allele frequencies (including disease alleles, which facilitates the application of personalized medicine).

MC.II.C. Sequencing of populations of RNA molecules (e.g., RNA-Seq) is a way to gain information about variation in which portions of the genome are expressed across individuals, environmental conditions, tissues, and cell types.

MC.III. Modern genomic technologies allow for the unbiased study of the metagenome.

MC.III.A. Metagenomics is the study of the community of microbial genomes and their expressed genes within a particular ecological niche (e.g., gut microbiome, soil).

MC.III.B. Metagenomics improves understanding of biological processes including human, animal, plant, and environmental health.

MC.IV. Rapid changes in technology have greatly improved the speed and accuracy of acquiring genomic data.

MC.IV.A. Technologies include improvements in DNA sequencing, RNA sequencing/mapping/quantification, protein identification and quantification, and mapping of chromatin and nuclear structure.

MC.IV.B. Each technology carries associated limitations and systematic biases that must be considered when analyzing genomic data.

MC.IV.C. Sequencing is never completely accurate and hence models of any particular species' genome may change with new data.

MC.IV.D. New technologies have made it possible to gather more types of genomic information from an increasingly wide range of species, and technological enhancements are expected to continue to advance rapidly.

MC.V. Computational algorithms assemble sequence data and generate predictions about the presence and structure of genes, molecular function of the gene products, and common ancestry of genomic regions.

MC.V.A. A well-studied and annotated genome of one species can act as a Reference Genome to facilitate the assembly and annotation of related species genomes.

MC.V.B. In some cases, manual analysis of the data can improve the accuracy of the annotations.

MC.V.C. Computational algorithms become more sophisticated with the ongoing addition of more types of and higher quality data, as well as improved computational power.

MC.V.D. Systems biology is the study of the interacting biological networks of both genomically encoded and environmentally determined factors.

MC.VI. An understanding of genomics facilitates the development of therapeutics, environmentally sustainable technology, and improved food security through biotechnology.

MC.VI.A. Genomic engineering through genome editing can create new technologies, enhance crop productivity, and provide disease treatments.

MC.VI.B. Genome editing technologies are improving in their accuracy and precision.

MC.VII. Genomics is an application of Data Science.

MC.VII.A. Computational considerations such as database structure and search algorithms are important.

MC.VII.B. Similar methodologies are also useful in other research fields.

**Ethical Concepts (EC)**

EC.I. Genomic information and technologies generate new ethical, medical, legal, and societal challenges and opportunities.

EC.I.A. Genomic information about people can be leveraged for good (such as curing a disease) or for ill (such as denying health insurance coverage).

| | |
|---|---|
| EC.I.B. Informed consent in the context of genomic data must be carefully considered. | EC.I.B.1. Genomic information is not just the property of an individual but also of any people that person is related to, including not only their close genetic relatives, but also their ethnic group or tribal affiliation. |

(*Continued*)

**Table 1.** (Continued)

| | |
|---|---|
| EC.I.C. Ethical frameworks for personalized medicine should be in place. | EC.I.C.1. Reproductive decisions may be impacted by genomic data. |
| | EC.I.C.2. The ability of the patient and medical professionals to accurately assess risk must be considered. |
| EC.I.D. Legal protection of individuals' genomic data on an international scale should be in place as that information could impact employment, health and life insurance, healthcare decisions, and law enforcement. | |
| EC.I.E. Genomic information may impact the social standing and/or psychological health of individuals and their communities. | EC.I.E.1. The relationship between genetic variation within humans, and how that variation is considered in the context of population and local politics must be acknowledged. |
| | EC.I.E.2. Individuals may be subjected to social stigmatization based on the genomic data. |
| | EC.I.E.3. Land-use rights may be influenced by genomic data. |
| | EC.I.E.4. Knowledge of one's own or a loved-one's genomic data may influence psychological health, such as revealing unexpected parentage or ancestry, or identifying future disease risks. |
| | EC.I.E.5. Knowledge of, or collection of, genomic data may potentially conflict with a group's cultural traditions such as creation stories or sanctity of the body. |
| | EC.I.E.6. There may be implications for individual and societal perceptions of free will versus genetic determinism. |
| EC.II. Genomic information and technologies may impact the environment and society. | |
| EC.II.A. Genomic information about the biosphere can be leveraged for good (such as preserving an endangered species) or for ill (such as introducing genetically-engineered traits into wild species with unintended consequences). | |
| EC.II.B. Genomic technologies can be leveraged to combat global challenges such as climate change, food insecurity, and pollution. | |
| EC.II.C. Genomic technologies may help to preserve biodiversity. | |
| EC.II.D. Considerations of how genetically modified organisms (GMOs) are to be regulated must be made. | |
| EC.II.E. Access to potential environmental and agricultural improvements enabled by genomic technologies should be shared equitably and globally. | |
| EC.II.F. Vulnerable groups and habitats should be protected from possible negative impacts of genomic technologies. | |
| EC.II.G Inequities between nations may fuel unjust exploitation of genomics resources obtained in areas beyond national jurisdiction (high seas) or foreign territories (biopiracy) through patent law for commercial purposes. | |

energy metabolism, and homeostasis, which are not features of the genomics concepts (S5–S7 Tables). Similarly, bioinformatics inventories showed intermediate levels of overlap, sharing ideas related to computation and algorithms, but not overlapping on specific genomic technologies or the biology of genome structure (S8–S10 Tables). In summary, the comparison to other inventories suggests that the genomics concepts fill a niche not fulfilled by existing inventories or learning frameworks.

## Discussion / Conclusion

McKusick and Ruddle [46] adopted, for "the newly developing discipline of mapping/sequencing (including analysis of the information)", the term *Genomics*. While in subsequent decades genomics has inserted itself into almost all aspects of life science research, even 30+ years later, instruction in genomics varies widely across different institutions. To help bring

**Table 2. Genomics concepts aligned with vision and change.**

| | VISION & CHANGE CORE CONCEPTS | | | | | VISION & CHANGE CORE COMPETENCIES | | | | | |
|---|---|---|---|---|---|---|---|---|---|---|---|
| | Evolution | Structure and Function | Information flow, exchange, and storage | Pathways and transformations of energy and matter | Systems | Apply the process of science | Use quantitative reasoning | Use modeling and simulation | Tap into the interdisciplinary nature of science | Communicate and collaborate with other disciplines | Understand the relationship between science and society |
| | *The diversity of life evolved over time by processes of mutation, selection, and genetic change.* | *Basic units of structure define the function of all living things.* | *The growth and behavior of organisms are activated through the expression of genetic information in context.* | *Biological systems grow and change by processes based upon chemical transformation pathways and are governed by the laws of thermodynamics.* | *Living systems are interconnected and interacting.* | *Biology is evidence based and grounded in the formal practices of observation, experimentation, and hypothesis testing.* | *Biology relies on applications of quantitative analysis and mathematical reasoning.* | *Biology focuses on the study of complex systems.* | *Biology is an interdisciplinary science.* | *Biology is a collaborative scientific discipline.* | *Biology is conducted in a societal context.* |
| **GENOMICS CONCEPTS** | | | | | | | | | | | |
| **Biological Concepts** | | | | | | | | | | | |
| BC.I. Evolutionary forces and processes inform our understanding and interpretation of genomic information. | Evolution is a key driver in living systems | | | | | | | | | | |
| BC.II. Genomes exhibit sequence variation (SNPs, etc.) and structural variation (rearrangements, copy number variation, transposable elements) within and between species that can lead to differences in form and function. | Comparisons within and between species is part and parcel of evolution | Genomes are made of genes and other elements, which are basic structural elements of life | Genomes contain genes which encode RNAs and proteins | | | | | Genomes are complex systems; modeling helps interpret the functions contained within a genome | | | |
| BC.III. Individual regions of a genome (e.g., promoters, enhancers, protein coding regions, non-coding RNAs etc.) are expected to have different functions as a result of sequence and structural differences. | | Individual regions of the genome are part of the structure of life. | The genome contains the sum of information expressed in different contexts. | | | | | | | | |

*(Continued)*

**Table 2.** (Continued)

| | VISION & CHANGE CORE CONCEPTS | | | | | VISION & CHANGE CORE COMPETENCIES | | | | | |
|---|---|---|---|---|---|---|---|---|---|---|---|
| | Evolution | Structure and Function | Information flow, exchange, and storage | Pathways and transformations of energy and matter | Systems | Apply the process of science | Use quantitative reasoning | Use modeling and simulation | Tap into the interdisciplinary nature of science | Communicate and collaborate with other disciplines | Understand the relationship between science and society |
| | *The diversity of life evolved over time by processes of mutation, selection, and genetic change.* | *Basic units of structure define the function of all living things.* | *The growth and behavior of organisms are activated through the expression of genetic information in context.* | *Biological systems grow and change by processes based upon chemical transformation pathways and are governed by the laws of thermodynamics.* | *Living systems are interconnected and interacting.* | *Biology is evidence based and grounded in the formal practices of observation, experimentation, and hypothesis testing.* | *Biology relies on applications of quantitative analysis and mathematical reasoning.* | *Biology focuses on the study of complex systems.* | *Biology is an interdisciplinary science.* | *Biology is a collaborative scientific discipline.* | *Biology is conducted in a societal context.* |
| BC.IV. Genomes contain information that determines both temporal and spatial patterns of gene expression and the response to environmental conditions. | | | The genome contains the sum of information expressed in different contexts. | Rates of RNA polymerization (transcription) and amino acid polymerization (translation) are limited by thermodynamics | Genomes of individuals, species, and groups determine interactions under given environmental conditions. | | | | | | |
| BC.V. Most cells within a multicellular organism host a nearly identical copy of the zygotic genome. | | The zygotic genome determines the behavior of individual cells in a multicellular organism and cells are a basic structure of life. | | Rates of DNA polymerization (genome replication) are limited by thermodynamics | | | | | | | |
| BC.VI. Metagenomics is the study of the community of microbial genomes and their expressed genes within a particular ecological niche (e.g. gut microbiome, soil) | Metagenomes are indicative of the diversity of life. | | Genes within metagenomes respond to environmental context. | | Organisms represented in metagenomes interact with one another, both positively and negatively. | | | | | Metagenomics requires information from a number of subdisciplines within Biology (microbiology, biochemistry) as well as statistics and computer science. | |
| **Methodological Concepts** | | | | | | | | | | | |

*(Continued)*

**Table 2.** (Continued)

| | VISION & CHANGE CORE CONCEPTS | | | | | VISION & CHANGE CORE COMPETENCIES | | | | | |
|---|---|---|---|---|---|---|---|---|---|---|---|
| | Evolution | Structure and Function | Information flow, exchange, and storage | Pathways and transformations of energy and matter | Systems | Apply the process of science | Use quantitative reasoning | Use modeling and simulation | Tap into the interdisciplinary nature of science | Communicate and collaborate with other disciplines | Understand the relationship between science and society |
| | *The diversity of life evolved over time by processes of mutation, selection, and genetic change.* | *Basic units of structure define the function of all living things.* | *The growth and behavior of organisms are activated through the expression of genetic information in context.* | *Biological systems grow and change by processes based upon chemical transformation pathways and are governed by the laws of thermodynamics.* | *Living systems are interconnected and interacting.* | *Biology is evidence based and grounded in the formal practices of observation, experimentation, and hypothesis testing.* | *Biology relies on applications of quantitative analysis and mathematical reasoning.* | *Biology focuses on the study of complex systems.* | *Biology is an interdisciplinary science.* | *Biology is a collaborative scientific discipline.* | *Biology is conducted in a societal context.* |
| MC.I. Genome-scale studies have statistical and experimental design considerations that impact their accuracy. | | | | | | Genomic experiments, like wet-lab experiments, involve observation and hypothesis testing. | Statistics and design involve quantitative reasoning. | Genomes are complex systems; modeling helps interpret the functions contained within a genome. | | | |
| MC.II. Within a species, genomic analyses provide information that permits deeper understanding of how the genome shapes an organism's phenotype | | | Which genes are expressed in a genome direct phenotype. | | | | Understanding of the extent to which genes are expressed influences phenotype. | | | | |
| MC.III. Modern genomic technologies allow for the unbiased study of the metagenome. | Metagenomes are indicative of the diversity of life. | | Genes within metagenomes respond to environmental context. | | Organisms represented in metagenomes interact with one another, both positively and negatively. | | Analyses of metagenome species composition requires application of quantitative ecological methods | | Acquiring metagenome level information is influenced and enhanced by advances in technology, including engineering and computer science. | Metagenomics requires information from a number of subdisciplines within Biology (microbiology, biochemistry) as well as statistics and computer science. | |
| MC.IV. Rapid changes in technology have greatly improved the speed and accuracy of acquiring genomic data. | | | | | | | | | Acquiring genome level information is influenced and enhanced by advances in technology, including engineering and computer science. | Technology changes influence ability to acquire genomic information. | |

*(Continued)*

**Table 2.** (Continued)

| | VISION & CHANGE CORE CONCEPTS | | | | | VISION & CHANGE CORE COMPETENCIES | | | | | |
|---|---|---|---|---|---|---|---|---|---|---|---|
| | Evolution | Structure and Function | Information flow, exchange, and storage | Pathways and transformations of energy and matter | Systems | Apply the process of science | Use quantitative reasoning | Use modeling and simulation | Tap into the interdisciplinary nature of science | Communicate and collaborate with other disciplines | Understand the relationship between science and society |
| | *The diversity of life evolved over time by processes of mutation, selection, and genetic change.* | *Basic units of structure define the function of all living things.* | *The growth and behavior of organisms are activated through the expression of genetic information in context.* | *Biological systems grow and change by processes based upon chemical transformation pathways and are governed by the laws of thermodynamics.* | *Living systems are interconnected and interacting.* | *Biology is evidence based and grounded in the formal practices of observation, experimentation, and hypothesis testing.* | *Biology relies on applications of quantitative analysis and mathematical reasoning.* | *Biology focuses on the study of complex systems.* | *Biology is an interdisciplinary science.* | *Biology is a collaborative scientific discipline.* | *Biology is conducted in a societal context.* |
| MC.V. Computational algorithms assemble sequence data and generate predictions about the presence and structure of genes, molecular function of the gene products, and common ancestry of genomic regions. | Common ancestry is a tenet of evolutionary theory. | Genes define a basic unit of life. | | | | Algorithms for interpreting genomic information involve observation and hypothesis testing. | Algorithms are based on statistical inferences, quantitative, and mathematical reasoning. | Algorithms are based on application of models. | | | |
| MC.VI. An understanding of genomics facilitates the development of therapeutics, environmentally sustainable technology, and improved food security through biotechnology. | | | | | Applications based on genomic information have the potential to influence living systems and their interactions. | Applying the information obtained through investigations of genomes involves developing new hypotheses. | | | Information obtained from genomics can lead to applications in other subdisciplines of Biology (protein structure and function, microbiology, cell biology, etc.) | Information obtained from genomics can lead to applications in other scientific areas, including Chemistry | Information obtained from genomics can be used to develop applications that improve the human condition. |
| MC.VI. Genomics is an application of Data Science. | | | | | | | Genomics requires quantitative reasoning and appropriate statistical approaches to deal with big data | Genomics involves the study of large amounts of data, leading to development of models and simulations to explain complex biological systems | An understanding of genomics requires some familiarity with data science modalities, including methods for understanding big data sets. | | |
| **Ethical Concepts** | | | | | | | | | | | |

*(Continued)*

**Table 2.** (Continued)

| | VISION & CHANGE CORE CONCEPTS | | | | | VISION & CHANGE CORE COMPETENCIES | | | | | |
|---|---|---|---|---|---|---|---|---|---|---|---|
| | Evolution | Structure and Function | Information flow, exchange, and storage | Pathways and transformations of energy and matter | Systems | Apply the process of science | Use quantitative reasoning | Use modeling and simulation | Tap into the interdisciplinary nature of science | Communicate and collaborate with other disciplines | Understand the relationship between science and society |
| | *The diversity of life evolved over time by processes of mutation, selection, and genetic change.* | *Basic units of structure define the function of all living things.* | *The growth and behavior of organisms are activated through the expression of genetic information in context.* | *Biological systems grow and change by processes based upon chemical transformation pathways and are governed by the laws of thermodynamics.* | *Living systems are interconnected and interacting.* | *Biology is evidence based and grounded in the formal practices of observation, experimentation, and hypothesis testing.* | *Biology relies on applications of quantitative analysis and mathematical reasoning.* | *Biology focuses on the study of complex systems.* | *Biology is an interdisciplinary science.* | *Biology is a collaborative scientific discipline.* | *Biology is conducted in a societal context.* |
| EC.I. Genomic information and technologies generates new ethical, medical, legal, and societal challenges and opportunities. | | | | | | | | | Genomic information exists within the interdisciplinary nature of science, including Biology. | | Genomic information needs to be considered within the context of society. |
| EC.II. Genomic information and technologies may impact the environment. | | | | | Changes to genomes as a consequence of technology (ex. CRISPR) might alter interactions between organisms and between organisms and the environment. | | | | Potential changes to genomes are a consequence of applications of other sciences to this subdiscipline of Biology. | | Genomic information and changes that might result needs to be considered within the context of society. |

undergraduate biology education into better alignment with the current practice of biological and biomedical research, we, a group of educators who teach genomics, have developed and codified a list of genomics concepts for use by other educators. As shown in the pairwise comparisons to inventories developed for other disciplines, this list of concepts is a novel contribution to biology education practices. Importantly, there is a strong alignment of this list of concepts with those laid out in the "Vision and Change in Undergraduate Biology Education" recommendations [19, 20], suggesting that genomics is an excellent way to deliver the goals of V&C to students. The distinctive overlap with concepts relevant to undergradute genetics, biochemistry, and bioinformatics education (shown in S3–S10 Tables) suggests that classical concepts in biology can be conveyed to undergraduates by aligning with or otherwise integrating genomics concepts.

Thus, we have created a new learning framework through input from a wide community of undergraduate genomics educators, one for genomics education, which we believe will be broadly useful to faculty who are designing coursework in this area. Our future goals will be to develop associated learning objectives and assessments for these concepts. Implementation of these goals for student instruction and faculty professional development will be informed by efforts to implement other inventories. Finally, as the field progresses, we predict that periodic revisions of the concepts will be necessary to keep current.

## Supporting information

**S1 File. Survey of undergraduate faculty teaching genomics.**
(PDF)

**S1 Fig. Relationship between class size and genomics content in courses taught by faculty participating in the inventory survey.** Introductory (A) and Upper-level (B) courses tended to contain more genomics when they were small, and class size tended to decrease in size at the upper relative to the introductory level.
(TIF)

**S2 Fig. Distributions of initial concept importance scores (1 = unimportant, 5 = extremely important) for biology majors to know as scored by faculty participating in the inventory survey.** Initial concepts are provided in S1 Table. A. Initial biological concept A and subparts (A1-A5). B. Initial biological concepts B-D and subparts (D1-D4). C. Initial biological concepts E and subparts (E1-E6). D. Initial methodological concepts A-G.
(TIF)

**S3 Fig. Concept importance variation and correlation for initial concept importance scores (1 = unimportant, 5 = extremely important) for biology majors to know as scored by faculty participating in the inventory survey.** B.# = biological concept, M.# = methodological concept, as listed in S1 Table. A. Average importance scores (+/- 1SE) in rank order from most to least important. All concepts were scored as being moderately to highly important (>3.73), with biological concepts generally ranked more highly. B. Correlations of importance scores for initial concepts clustered based on correlation. Methodological concept scores tended to correlate with other methodology concept scores, and biological importance scores tended to correlate with other biological concept scores.
(TIF)

**S4 Fig. Frequency and proportion of initial genomic concepts were taught in courses lead by faculty participating in the inventory survey.** Initial concepts are provided in S1 Table. All biological concepts were taught in at least some genomics courses, but they were taught

more frequently at the upper level (A). When normalized to the number of faculty teaching any genomics content at the introductory or upper level, only biological concepts A.B and A.D were taught in a significantly larger proportion of the upper-level courses (B). All methodological concepts were taught in at least some genomics courses, but they were taught more frequently at the upper level (C). Methodological concepts were also taught in a significantly higher proportion of the upper-level courses when normalized to the number of courses taught, except for concept M.G (D).* $p < 0.05$, ** $p < 0.01$, *** $p < 0.001$, **** $p < 0.0001$ by comparison of two proportions Z-test.
(TIF)

**S1 Table. Initial concepts.**
(XLSX)

**S2 Table. Comparison of Initial to final concepts.**
(XLSX)

**S3 Table. Genomics concepts aligned with genetics concepts learning goals.**
(XLSX)

**S4 Table. Genomics concepts aligned with CourseSource genetics framework.**
(XLSX)

**S5 Table. Genomics concepts aligned with foundational concepts in biochemistry and molecular biology.**
(XLSX)

**S6 Table. Genomics concepts aligned with threshold concepts for biochemistry.**
(XLSX)

**S7 Table. Genomics concepts aligned with the CourseSource biochemistry and molecular biology framework.**
(XLSX)

**S8 Table. Genomics concepts aligned with the NIBLSE core competencies.**
(XLSX)

**S9 Table. Genomics concepts aligned with the mastery rubric for bioinformatics.**
(XLSX)

**S10 Table. Genomics concepts aligned with the CourseSource bioinformatics framework.**
(XLSX)

## Acknowledgments

The first iterations of the Genomics Concepts were articulated and discussed at a Genomics Education Alliance workshop in June 2019. We thank our colleagues who participated at that meeting, including Cindy Arrigo (New Jersey City University), Judy Brusslan (California State University Long Beach), Sam Donovan (BIOQuest), Sarah Elgin (Washington University in St. Louis), Matt Escobar (California State University San Marcos), Alexa Sawa (College of the Desert), Naomi Stover (Illinois State University), and Emily Wiley (Claremont McKenna College). We also thank those who participated in the 2020 BIOME Conference for their input as well as the individuals who completed the survey.

We dedicate this paper to our colleague, Vince Buonoccorsi, a tireless advocate for undergraduate education who passed away too soon.

## Author Contributions

**Conceptualization:** Laura K. Reed, Adam J. Kleinschmit, Vincent Buonaccorsi, Arthur G. Hunt, Douglas Chalker, Jason Williams, Christopher J. Jones, Juan-Carlos Martinez-Cruzado, Anne Rosenwald.

**Formal analysis:** Laura K. Reed, Anne Rosenwald.

**Funding acquisition:** Vincent Buonaccorsi, Douglas Chalker, Jason Williams, Anne Rosenwald.

**Investigation:** Adam J. Kleinschmit.

**Methodology:** Adam J. Kleinschmit, Vincent Buonaccorsi, Anne Rosenwald.

**Visualization:** Laura K. Reed.

**Writing – original draft:** Laura K. Reed, Adam J. Kleinschmit, Vincent Buonaccorsi, Arthur G. Hunt, Douglas Chalker, Jason Williams, Christopher J. Jones, Juan-Carlos Martinez-Cruzado, Anne Rosenwald.

**Writing – review & editing:** Laura K. Reed, Adam J. Kleinschmit, Arthur G. Hunt, Douglas Chalker, Jason Williams, Christopher J. Jones, Juan-Carlos Martinez-Cruzado, Anne Rosenwald.

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
