## [Decision Letter · Decision Letter 0]

23 Jul 2024

PONE-D-24-14148A genomics learning framework for undergraduatesPLOS ONE

Dear Dr. Rosenwald,

Thank you for submitting your manuscript to PLOS ONE. After careful consideration, we feel that it has merit but does not fully meet PLOS ONE’s publication criteria as it currently stands. Therefore, we invite you to submit a revised version of the manuscript that addresses the points raised during the review process.

Please review suggestions and questions from the reviewers and address them in your response.

We look forward to receiving your revised manuscript.

Kind regards,

Surya Saha, PhD

Academic Editor

PLOS ONE

Journal Requirements:

"The efforts of the Genomics Education Alliance were supported by the National Science Foundation DBI 1827130."

"The author(s) received no specific funding for this work"

6. Please include your tables as part of your main manuscript and remove the individual files. Please note that supplementary tables (should remain/ be uploaded) as separate ""supporting information"" files

Reviewers' comments:

Reviewer's Responses to Questions

**Comments to the Author**

1. Is the manuscript technically sound, and do the data support the conclusions?

Reviewer #1: Yes

Reviewer #2: Yes

2. Has the statistical analysis been performed appropriately and rigorously? 

Reviewer #1: Yes

Reviewer #2: Yes

3. Have the authors made all data underlying the findings in their manuscript fully available?

Reviewer #1: Yes

Reviewer #2: Yes

4. Is the manuscript presented in an intelligible fashion and written in standard English?

Reviewer #1: Yes

Reviewer #2: Yes

5. Review Comments to the Author

Reviewer #1: The manuscript is well conceived and clearly written. The authors have created a thorough list of concepts for genomics that will be very helpful to educators. I have only a few minor suggestions.

The list of methodological concepts mentions genome sequencing and metagenomics, but does not mention RNA-Seq. Adding a concept (perhaps MC.II.C) about RNA-Seq as a way to gain information about which portions of the genome are expressed and to compare expression under different conditions/cell types would make the methodological concepts more complete. However, since that would require redoing all of the comparisons to other inventories, adding an explicit mention of RNA-Seq to BC.IV.C would probably suffice.

In concept EC.II.G Should it be “patent” law instead of “patient” law?

Table S4: The Course Source Genetics Core Concepts Headings need to be wrapped so they are completely visible

Reviewer #2: The manuscripts presents the development of a genomics concept list that provides an updated set of topics and subjects that should be considered in development and instruction of genomics courses. The development and design of this concepts list was thorough and comprehensive, with valuable input by appropriate faculty. The authors provide detailed comparison with existing lists and inventories to demonstrate that there is a niche and need for this updated concept list. Further, authors mention how the new concept list will be useful in development of genomics curriculum and assessment.

The manuscript is organized well, methodology is clearly stated and the information is clearly presented.

Specific areas to review:

The supplementary figures were not made available. The supplementary figure legends were provided and the main result of the figures was presented in the text. However, the figures were not available for review.

Line 79: PHAGES is acronym, should be capitalized and hyphenated

Line 147: A short description of the genomic educators would be helpful, for instance what level courses (generally lower/upper level) and/or what type of institutions (2 yr, 4 yr, grad?) do they represent? Are they the n=15 BIOME members from line 203?

Line 148-151: how many faculty were surveyed

Table S1: “rekecting” in B.A. section

Paragraph 282-296: Consider: It could be helpful to point out some of the examples where the genomics concepts do not overlap the existing inventories, indicating support for the newly developed genomics concepts list. At least for the comparison to bioinformatics inventories (line 293), what was missing from existing bioinformatics inventories that this new genomics concept list offers? To a reader this would provide a quick indication of how, in addition to V&C and other inventories, the presented genomics concepts provides an updated and more complete set of concepts relating to instructing genomics.

6. PLOS authors have the option to publish the peer review history of their article (what does this mean?). If published, this will include your full peer review and any attached files.

Reviewer #1: No

Reviewer #2: No

---

## [Author Response · Author response to Decision Letter 0]

4 Sep 2024

Response to reviewers

PONE-D-24-14148

A genomics learning framework for undergraduates

We examined these two links and modified the manuscript accordingly. 

Participants were informed of the nature of the survey at the beginning of the survey. They could leave the survey at any time without penalty. Thus, completion of the survey was implicit consent. Moreover, the survey was subjected to IRB review at both Georgetown University and Washington University in St. Louis and was deemed exempt at both institutions. 

When we began this work, the Genomics Education Alliance was supported by an REU grant from the National Science Foundation (Grant # 1827130 from the Division of Biological Infrastructure [DBI]). However, the grant expired in 2022. Thus, the final stages of this work were not supported by any grant. Please advise as to the best way to report this information. 

"The efforts of the Genomics Education Alliance were supported by the National Science Foundation DBI 1827130."

Funding information has been removed from the Acknowledgments section.

The statement has been removed as these data are not a core part of the study. What we meant by this is we performed the matchup as in Table 2 and the Supplementary Tables, but there were few places where these alternate inventories matched with ours, which is essentially how we report this information now. We agree that the statement “data not shown” is superfluous in these instances. 

6. Please include your tables as part of your main manuscript and remove the individual files. Please note that supplementary tables (should remain/ be uploaded) as separate ""supporting information"" files.

We have added Tables 1 and 2 to the main manuscript. 

We have reviewed the references. None of the papers have been retracted. 

Reviewers' comments:

Reviewer's Responses to Questions

Comments to the Author

1. Is the manuscript technically sound, and do the data support the conclusions?

Reviewer #1: Yes

Reviewer #2: Yes

2. Has the statistical analysis been performed appropriately and rigorously?

Reviewer #1: Yes

Reviewer #2: Yes

3. Have the authors made all data underlying the findings in their manuscript fully available?

The PLOS Data policy<http://www.plosone.org/static/policies.action#sharing> requires authors to make all data underlying the findings described in their manuscript fully available without restriction, with rare exception (please refer to the Data Availability Statement in the manuscript PDF file). The data should be provided as part of the manuscript or its supporting information, or deposited to a public repository. For example, in addition to summary statistics, the data points behind means, medians and variance measures should be available. If there are restrictions on publicly sharing data—e.g. participant privacy or use of data from a third party—those must be specified.

Reviewer #1: Yes

Reviewer #2: Yes

4. Is the manuscript presented in an intelligible fashion and written in standard English?

Reviewer #1: Yes

Reviewer #2: Yes

5. Review Comments to the Author

Reviewer #1: The manuscript is well conceived and clearly written. The authors have created a thorough list of concepts for genomics that will be very helpful to educators. I have only a few minor suggestions.

The list of methodological concepts mentions genome sequencing and metagenomics, but does not mention RNA-Seq. Adding a concept (perhaps MC.II.C) about RNA-Seq as a way to gain information about which portions of the genome are expressed and to compare expression under different conditions/cell types would make the methodological concepts more complete. However, since that would require redoing all of the comparisons to other inventories, adding an explicit mention of RNA-Seq to BC.IV.C would probably suffice.

Thank you for noticing this issue. We have added MC.II.C “Sequencing of populations of RNA molecules (e.g., RNA-Seq) is a way to gain information about variation in which portions of the genome are expressed across individuals, environmental conditions, tissues, and cell types.” This addition has minimal impact on the concept inventory comparisons and the appropriate updates have been made.

In concept EC.II.G Should it be “patent” law instead of “patient” law?

Yes! We have corrected this typo. Thank you for catching it.

Table S4: The Course Source Genetics Core Concepts Headings need to be wrapped so they are completely visible

These headings have been adjusted for visibility.

Reviewer #2: The manuscripts presents the development of a genomics concept list that provides an updated set of topics and subjects that should be considered in development and instruction of genomics courses. The development and design of this concepts list was thorough and comprehensive, with valuable input by appropriate faculty. The authors provide detailed comparison with existing lists and inventories to demonstrate that there is a niche and need for this updated concept list. Further, authors mention how the new concept list will be useful in development of genomics curriculum and assessment.

The manuscript is organized well, methodology is clearly stated and the information is clearly presented.

Specific areas to review:

The supplementary figures were not made available. The supplementary figure legends were provided and the main result of the figures was presented in the text. However, the figures were not available for review.

We apologize for this oversight. The supplementary figures were part of the same file containing Figure 1 but was not labeled adequately. All figures are now available as individual tiff files. 

Line 79: PHAGES is acronym, should be capitalized and hyphenated

This correction has been made.

Line 147: A short description of the genomic educators would be helpful, for instance what level courses (generally lower/upper level) and/or what type of institutions (2 yr, 4 yr, grad?) do they represent? Are they the n=15 BIOME members from line 203?

We added the phrase at the end line 148 “.....a combination of 2-year, 4-year, and research intensive institutions, both public and private”. The founding members of GEA and others who attended our workshops and virtual meetings as we developed these genomics concepts included four from community colleges, five from private, 4-year residential colleges, and the remainder from public or private research-intensive (R1 or R2) institutions. We considered both introductory and advanced undergraduate curricula. The 15 BIOME members provided additional feedback during the 2020 virtual workshop before we surveyed a larger group of educators as described.

Line 148-151: how many faculty were surveyed

61, this has been added to the text

Table S1: “rekecting” in B.A. section

This has been edited to say “reflecting”. Thank you for noticing the typo.

Paragraph 282-296: Consider: It could be helpful to point out some of the examples where the genomics concepts do not overlap the existing inventories, indicating support for the newly developed genomics concepts list. At least for the comparison to bioinformatics inventories (line 293), what was missing from existing bioinformatics inventories that this new genomics concept list offers? To a reader this would provide a quick indication of how, in addition to V&C and other inventories, the presented genomics concepts provides an updated and more complete set of concepts relating to instructing genomics.

We have chosen not to include examples in the text given that the places where overlap is not seen are given in the Tables (both Table 2 and the Supplementary Tables). 

6. PLOS authors have the option to publish the peer review history of their article (what does this mean?<https://journals.plos.org/plosone/s/editorial-and-peer-review-process#loc-peer-review-history>). If published, this will include your full peer review and any attached files.

Do you want your identity to be public for this peer review? For information about this choice, including consent withdrawal, please see our Privacy Policy<https://www.plos.org/privacy-policy>.

Reviewer #1: No

Reviewer #2: No

---

## [Decision Letter · Decision Letter 1]

21 Oct 2024

A genomics learning framework for undergraduates

PONE-D-24-14148R1

Dear Dr. Rosenwald,

Congratulations!!

We’re pleased to inform you that your manuscript has been judged scientifically suitable for publication and will be formally accepted for publication once it meets all outstanding technical requirements. Thank you for your patience with the peer review process.

Kind regards,

Surya Saha, PhD

Academic Editor

PLOS ONE

Additional Editor Comments (optional):

Reviewers' comments:

Reviewer's Responses to Questions

**Comments to the Author**

1. If the authors have adequately addressed your comments raised in a previous round of review and you feel that this manuscript is now acceptable for publication, you may indicate that here to bypass the “Comments to the Author” section, enter your conflict of interest statement in the “Confidential to Editor” section, and submit your "Accept" recommendation.

Reviewer #1: All comments have been addressed

Reviewer #2: All comments have been addressed

2. Is the manuscript technically sound, and do the data support the conclusions?

Reviewer #1: Yes

Reviewer #2: Yes

3. Has the statistical analysis been performed appropriately and rigorously? 

Reviewer #1: Yes

Reviewer #2: Yes

4. Have the authors made all data underlying the findings in their manuscript fully available?

Reviewer #1: Yes

Reviewer #2: Yes

5. Is the manuscript presented in an intelligible fashion and written in standard English?

Reviewer #1: Yes

Reviewer #2: Yes

6. Review Comments to the Author

Reviewer #1: (No Response)

Reviewer #2: Authors have sufficiently provided the recommended revisions.

7. PLOS authors have the option to publish the peer review history of their article (what does this mean?). If published, this will include your full peer review and any attached files.

Reviewer #1: No

Reviewer #2: No

---

## [Editor Report · Acceptance letter]

11 Nov 2024

PONE-D-24-14148R1 

PLOS ONE

Dear Dr. Rosenwald, 

I'm pleased to inform you that your manuscript has been deemed suitable for publication in PLOS ONE. Congratulations! Your manuscript is now being handed over to our production team.

Kind regards, 

on behalf of

Dr. Surya Saha 

Academic Editor

PLOS ONE